# The Impact of Bending on Radiation Characteristics of Polymer-Based Flexible Antennas for General IoT Applications

Muhammad Usman Ali Khan [1,2,*], Raad Raad [1], Faisel Tubbal [1] and Panagiotis Ioannis Theoharis [1]

1 School of Electrical Computer and Telecommunication Engineering, University of Wollongong, Wollongong, NSW 2522, Australia; raad@uow.edu.au (R.R.); faisel@uow.edu.au (F.T.); pit289@uowmail.edu.au (P.I.T.)
2 Department of Electronic Engineering, The Islamia University of Bahawalpur, Bahawalpur 63100, Pakistan
* Correspondence: muak803@uowmail.edu.au or usman.ali@iub.edu.pk; Tel.: +61-405817010

**Abstract:** Flexible wearable wireless devices have found practical uses as their cost has fallen and Internet of Things applications have gained further acceptance. These devices are gaining further use and acceptance in the consumer and wearable space for applications such as logistical tracking and maintaining sensor information, including temperature, humidity, and location. In such applications, antennas are exposed to bending and crumbling. Therefore, flexible substrate antennas for use with polymer-based flexible devices are an important area of research that needs to be addressed. In this study, the bending capabilities of flexible polymer substrate antennas for general IoT applications were practically analyzed by fabricating flexible antennas on Polyethylene Terephthalate (PET), Polytetrafluoroethylene (PTFE) Teflon, and Polyvinylchloride (PVC) substrates operating at 2.45, 4.45, and 7.25 GHz frequencies. The basic premise was to investigate the flexibility and bending ability of polymer materials, and their tendency to withstand deformation. In the current paper, we start by providing an equivalent model for the flexible microstrip patch antenna under bent conditions, followed by outlining the process of designing flexible antennas on polymer substrates. Finally, the fabricated flexible antennas were tested in an anechoic chamber for various radiation characteristics such as reflection coefficients, operating frequency shifts, and impedance mismatch with the transmission line, under bending conditions up to 7 mm. The practical outcomes were then compared with our recent investigation on flexible polymer substrate antennas for wearable applications. This study provides a means to select a suitable polymer substrate for future wearable sensors and antennas with high bendability.

**Keywords:** polymers substrates; wearable sensors; flexible antennas; bending analysis; wearable applications

## 1. Introduction

Flexible wearable wireless devices are gaining significant attention because of their characteristics of light weight and low cost, low power consumption, high flexibility combined with robustness, and compact size. These wearable devices have become essential for many wireless applications. In recent decades, flexible sensors such as antennas and RFID tags have been designed for WLAN, GPS, military, and biomedical related applications [1–5]. Presently, as a result of the exponential development of wearable sensors and devices, and the high demand for the flexible electronic systems, various new challenges have arisen because of the unconventional performance requirements. Traditional antennas are customarily made of conductive wires or by etching metal patterns on rigid substrates. When subjected to stretching, folding, or twisting, these types of antennas become permanently deformed, if not broken, which renders them incompatible for applications that require high flexibility/bendability and are subject to this continuous deformation. Therefore, it is very important for the flexible sensors to be lightweight, small,

durable, moist and heat resistant, and, most importantly, highly flexible without distorting radiation characteristics.

Flexible antennas have gained significant attention in recent years due to their advantage of high flexibility, which directly addresses the above problem, in addition to their convenient integration with other microwave components [6], light weight, energy efficiency, reduced fabrication complexity, easy mountability on conformal surfaces, low cost, and abundant availability in the form of substrate films [7]. For this reason, this investigation focuses on the flexibility and seamless integrity of flexible antennas for wearable/IoT applications.

To achieve the aforementioned characteristics for flexible antennas, conventional conductors and substrate materials such as metals and ceramics are not essentially appropriate. This is because these materials are usually rigid, costly, and lack flexibility and mechanical resilience. A large amount of research has previously explored numerous materials that exhibit suitable properties as a substrate for conductive materials for flexible antennas, including conductive polymers, conductive threads, and conductive textile. For dielectric materials, Polyimide (PI), Polyethylene Terephthalate (PET), Polydimethylsiloxane (PDMS), Polytetrafluoroethylene (PTFE), and Liquid Crystal polymers (LCPs) have been explored. A detailed review of these conductive and polymer-based substrate materials is presented in our recent study in [1].

Wearable devices, such as flexible antennas, and RFID sensors and tags, are critically affected by bending, twisting, and crumpling [1,3,8,9]. These deformative actions are sometimes inevitable, resulting in detrimental changes in the radiation performance of the flexible device. To ensure the robustness and durability of these flexible devices under flex conditions, a few important parameters for flexible devices are required to be tested for performance, such as shifts in resonant frequency, return loss, and signal degradation [8,9]. Although radiation patterns, directivity, and gain are all factors that can be affected by bending [10], one of the major impacts of bending is on the resonant frequency and return losses, because the curvature on the device is prone to shift the resonant frequency either towards higher components or lower components of the resonant frequency and cause a change in its signal strength. The process of applied bending is predisposed to creating a mismatch in the impedance of the Transmission Line (TL) and the feedline, and changing the capacitance between the resonators and causing a modification in the effective length of the radiation element [10]. Therefore, the most reliable flexible antennas are considered to be those that tolerate the application of a certain level of bending to limit the adverse impacts on their radiation efficiency. In this regard, polymers are considered to be one of the best candidates as a flexible substrate for wearable applications [1]. For this reason, bending impacts on polymer-based flexible antennas, which include deviations from the central frequency, Transmission Line (TL) mismatch, and the reductions in signal strength were the focus for analysis and assessment in this study.

The remainder of the paper is organized as follows: Section 2 provides a tutorial on the flexible antenna structure, bending capabilities, and impacts on antenna parameters. The steps used to design flexible antennas are described in Section 3, including material selection, property examination, design and fabrication, and measurement and testing of flexible polymer-based antennas. Radiation characteristics and bending analysis are presented in Section 4, and the conclusions are presented in Section 5.

## 2. Transmission Line Equivalent Circuit Model of a Flexible Antenna

The major aim of this study was to investigate the capability of antennas to flex and function properly under bending conditions. When deformation of an antenna implicates a significant change in radiation characteristics, such as return loss of antennas, which results in a shift in the resonant frequency, the signal degradation is sometimes too high to allow the signal to be received appropriately at the receiving ends. Therefore, a Transmission Line (TL) model under flex conditions was analyzed to corroborate the impact of bending on the radiation characteristics of a flexible antenna.

In the case of flexible polymer-based antennas, the return loss is severely impacted when antennas undergo a certain level of bending. A simple TL model for a microstrip patch antenna is presented in Figure 1, where a co-planar microstrip circuit with aspect ratio $W_m/h$ is presented with an open-end termination where $W$, $L$, $Y_c$ and $W_m$, $L_m$, $Y_{cm}$ are width, length, and characteristic admittances for the microstrip patch and TL, respectively, $h$ is the height of the substrate, and $Ys$ is the self-admittance or radiation admittance and represents the open-ended TL of the microstrip antenna. The general TL model for a microstrip patch antenna is shown in Figure 1. In the case of the microstrip feed line, the equivalent is represented by an open-ended admittance $Y_S$.

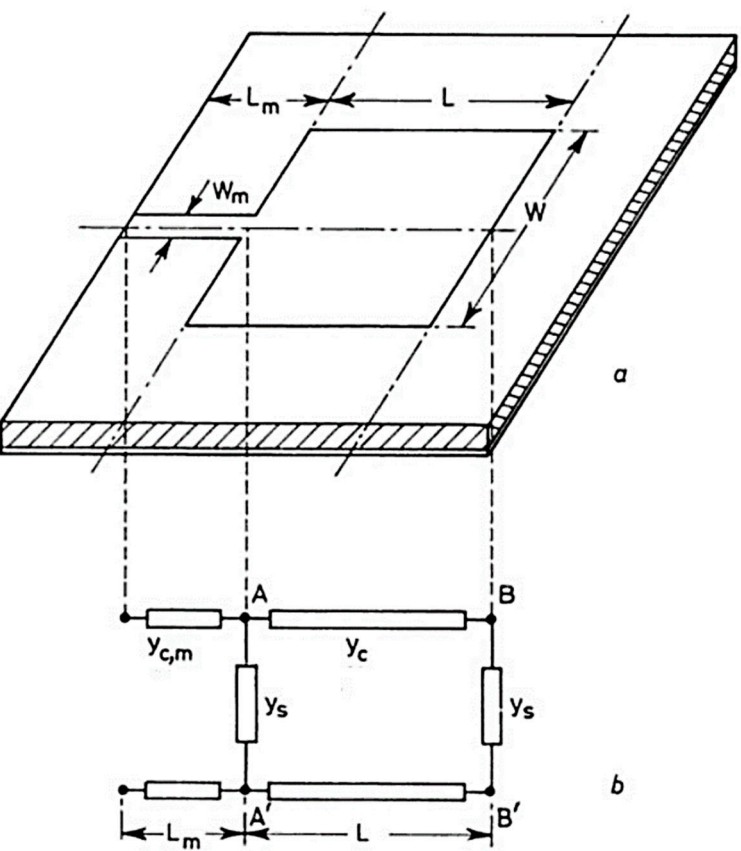

**Figure 1.** The TL model of a flexible microstrip antenna.

*Bending Impacts on the Parameters and Radiation Characteristics of Flexible Antenna*

Figure 2 illustrates the bending characteristic of the substrate at various bending levels. The bending of the antenna will change the antenna geometry and shape. Generally, bending causes a change in the effective length, thus introducing a change in the impedance characteristics of the antenna (i.e., mainly capacitance), causing a shift in the resonance frequency and a potential shift in the radiation pattern. Discontinuity in the shape or geometry of the antenna mainly leads to a distortion of the electric field between the patch and ground plane [11–14]. The model in Figure 2 describes the bending levels with additional capacitance, where the length L of the patch is broken up into *n* equal segments. This capacitance $C_{bend}$ is an additional capacitance from the bend of the antenna and varies with the bending curvature, and $\gamma$ is the propagation constant.

Previous studies have investigated the bending and crumbling effects of different antennas, but not necessarily polymer-based ones. The effects of bending on electromagnetic radiation performance have been addressed in the literature [6,15–17]. At its desired frequency, an antenna should, ideally, absorb all available energy when receiving and purge all its energy when transmitting. Under bent and flat conditions, according to [15], very small differences between input return losses are observed. The radiation pattern, the

distribution of the electrical field that propagates from the radiating element of the antenna, is an important characteristic that is mostly dependent on its geography. The crumpling of the antenna may adversely affect the radiation pattern. In [16], the authors present a dual-band coplanar waveguide antenna fabricated on a 48 × 33 mm polyimide substrate that operates at the frequencies of 2.45 and 5.75 GHz. Results showed that crumpling demonstrated a negligible impact on the antenna's performance, particularly regarding resonant frequency and gain, and illustrated that up to 5.3 mm crumpling curvature causes only minor changes in the resonant frequency while yielding a high gain. Moreover, this antenna can be integrated into flexible electronic devices that operate in multiple frequency bands.

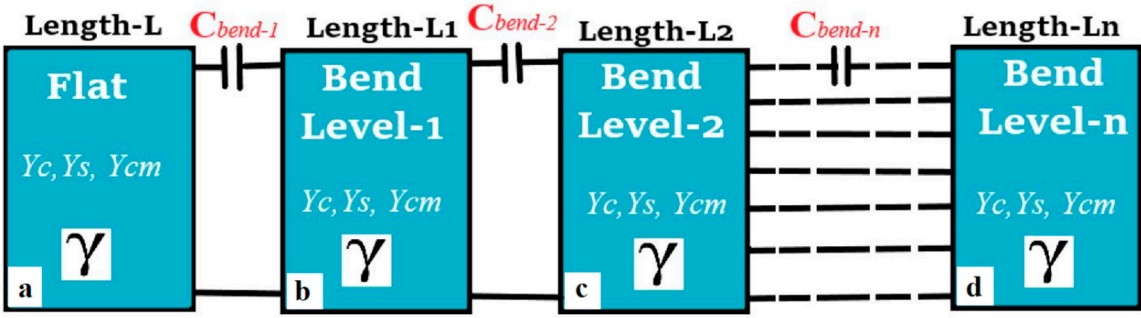

**Figure 2.** Antenna Equivalent Model for bent conditions, with circuit parameters Yc, Ys, Ycm (**a**) flat antenna, (**b**) Level 1 presents a minor radial bending, (**c**) Level 2 is higher, and (**d**) Level n is the highest level of curvature.

Different flexible materials that improve a bent antenna's efficiency and radiation characteristics have been considered in various studies and approaches [18]. Essentially, a bent antenna's performance is dependent on the electrical, physical, mechanical, and chemical properties of the flexible substrate material, which we stated comprehensively in [1]. These properties can be adjusted by the combination of different flexible materials [19]. In [6], the proposal of a UWB flexible antenna for flexible applications observed that a flexible substrate should consequently have high flexibility and robustness, and exhibit high tolerance against twisting and bending [6]. For such flexible antennas, the fabrication process may be a challenge because the properties of the flexible materials are significantly different from the materials used in the fabrication of traditional antennas. In the following section, the fabrication techniques that have been used extensively for flexible antenna designs are described.

## 3. Flexible Substrate Antenna Design

The design procedure for the development of the flexible substrate antennas, illustrated in Figure 3, is comprised of five major steps. First, the preferred conductive and substrate materials are selected according to the required application, in this case for a wearable flexible antenna. The selection of the substrate material for wearable applications is generally challenging because of the necessity for compatibility with the human body, and the need to be able to withstand twisting and bending without distorting its radiation characteristics. The second step is to investigate the electrical, mechanical, and thermal properties of the proposed dielectric flexible substrate in terms of its conductivity, resistivity of the conductive material and loss tangent, permittivity, dielectric strength, tensile modulus, CLTE, density, and moisture absorption, because a suitable flexible substrate needs to possess a high level of deformability, and thermal and electrical stability, with low fabrication complexity. Once the conductive and the substrate material are selected for the antenna, the next step is to design the antenna. This includes the designation of a geometrical shape appropriate for the required application for its frequency requirement, its mathematical modelling, which includes a dimensional and parametric analysis, and the simulation and optimization of the design. In the fourth step, an appropriate fabrication technique, based on the materials, is selected for the reproduction of the design on the

flexible substrates. Finally, the fabricated flexible antenna is tested for various radiation characteristics such as reflection coefficients, radiation patterns, gain, and directivities of the antenna. Further qualitative tests for wearable applications, irrelevant to conventional antennas on rigid substrates, such as bending impact on *S*-parameters, durability, robustness, moisture, and thermal tests, are also necessary.

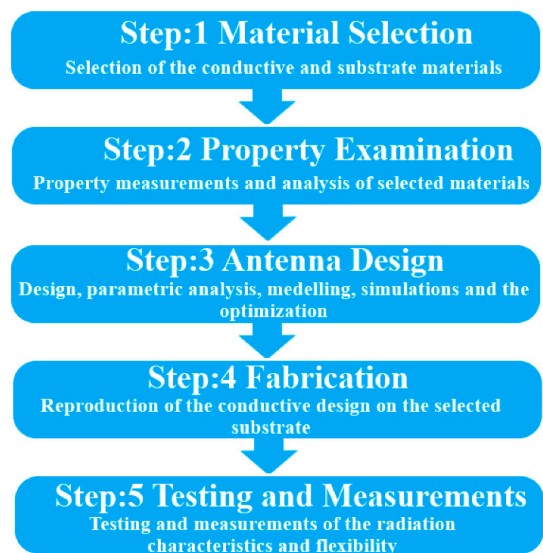

**Figure 3.** Five steps in the design procedure for a flexible substrate antenna.

Significant research has been conducted on polymers in recent decades for their application in flexible antennas, due to their advantages of high flexibility, convenient integration with other microwave components [6], light weight, energy efficiency, reduced fabrication complexity, easy mountability on conformal surfaces, low cost, and abundant availability in the form of substrate films [7]. In this study, attention was dedicated to the design and investigation of flexible polymer-based wearable antennas. The different steps for the flexible polymer-based antenna design procedure are described below.

*3.1. Material Selection*

To comply with the flexibility requirements of the antennas, the materials selected for the conductor, in general, and the substrate more specifically, need to be highly flexible and mechanically robust. Detailed analysis on the conductive and substrate materials was demonstrated in [1], which describes the range of properties for suitable polymer substrate materials. In the case of the conductive material, which is the ground plane and radiating element, copper is selected due to its qualities of being highly conductive and possessing very low resistivity. Moreover, copper is highly flexible, withstands a high level of bending and crumpling, has high tensile strength with an ability to withstand repeated pressure and deformations, and is resistant to corrosion and oxidization. See Table 1 for the properties of pure copper.

**Table 1.** Properties of pure copper.

| Properties of Pure Copper [20] | | | | | | | |
|---|---|---|---|---|---|---|---|
| **Physical** | | **Electrical** | | **Mechanical** | | **Thermal** | |
| Melting Point | Density | Conductivity | Temperature Coefficient | Tensile Strength | Modulus | Thermal Conductivity | CTE |
| 1083 °C | 8.96 g/cm$^3$ at 20 °C | 58 MS/m at 20 °C | 0.0043/K at 0–100 °C | 224–314 MPa | 137.8 | 401 W/m-K at 0–100 °C | $17.0 \times 10^{-6}$ m/m-K at 0–100 °C |

The selection of an appropriate polymer substrate is very important for flexible antennas. It should be highly flexible, easy to integrate with a conductive material, have a high tolerance level for bending or crumpling, and have a high repeatability endurance. To corroborate the impact of curvature on different polymer substrates, PET (Melinex 401), PTFE Teflon, and flexible PVC were chosen. The selection of these three dielectric polymers was based on the analysis of their physical, electrical, and thermal properties, which showed their suitability for bending applications [1,21].

### 3.2. Property Examination

After the selection of the conductive and substrate materials, the next step is to analyze the properties of the selected materials. The propagation and loss properties of the dielectric substrate need to be known for the prospective material before it is implemented as the substrate of the antenna. For this purpose, the properties of the conductive material and the polymers—PET, PTFE Teflon, and PVC sheets—are listed in Table 2. Results indicate that the PTFE Teflon sheet with a thickness of 100 μm has the highest density, temperature CTE, and shrinking capability, at 150° C, and the lowest dielectric strength, dissipation factor, and moisture absorption capability. This establishes Teflon as a good choice for a moisture- and thermal-resistant polymer material. The next substrate, PET with a thickness of 70 μm, possesses good physical and electrical properties, such as high tensile and dielectric strength and low dissipation factor, making it suitable for highly deformational applications.

**Table 2.** Properties of flexible polymers: PET, PTFE, PVC.

| Substrate | Model/ Thickness | Physical/Mechanical Properties | | | Electrical Properties | | | Thermal/Chemical Properties | | |
|---|---|---|---|---|---|---|---|---|---|---|
| | | Density (g/cc) | Tensile Strength X-Direction at 23 °C (Kpsi) | Tensile Modulus X-Direction at 23 °C (Kpsi) | Dielectric Constants 100 Hz to 1 GHz | Dielectric Strength (V/mil) | Dissipation Factor Tan σ at 100 Hz to 1 GHz | CTE-15 °C to 300 °C (ppm/C) | Moisture Absorption (%) at 23 °C | Shrinkage (%) 30 min, 150 °C |
| **PET** | Melinex 401 Polyester 70 μm | 1.3 | 25 | 420 | 2.07 | 4000 | 0.002 | 19 to 20 | 0.1 to 0.7 | 0.5 to 1.1 |
| **PTFE** | Teflon 100 μm | 2.1 | 3.9 | 65 | 2.70 | 285 | 0.0002 | 250 to 275 | 0 to 0.05 | 1.5 to 3.0 |
| **PVC** | Flexible PVC-O 110 μm | 1.4 | 2.2 | 217 | 3.70 | 635 | 0.04 | 6 to 7 | 0.2 to 1 | 0.2 to 2 |

Flexible PVC is a low-cost material with high strength, and is formed by the addition of compatible plasticizers to PVC to lower the crystallinity of the material. It has very good resistance to ultraviolet (UV) radiation and is non-reactant to various organic materials, such as crude oil at high temperatures. Comparatively, it has the worst physical properties, such as the lesser ability to withstand high pressure and deformability. Hence, these three polymer substrates, in terms of their ability to withstand high bending, can be placed in the order of PET, then PTFE, and PVC in the lowest position. Classification in terms of thermal properties would resort this order to PVC, then PET, and then PTFE, with Teflon at the lowest position. The implications of certain curvatures on these materials may result in additional impacts that influence the use of one of these polymers as the selected substrate for the antennas. As previously discussed in the first part of this article, the bending of the antenna with a flexible polymer substrate has the greatest impact on its resonant frequency, *S*-parameters, and radiation efficiency. Figure 4 shows the actual PET, Teflon, and PVC polymer sheets used as the substrates in fabricating our proposed flexible antenna design. After the properties of the polymer substrates and the conductive and substrate material are tested, the next step is to develop the antenna's shape and design.

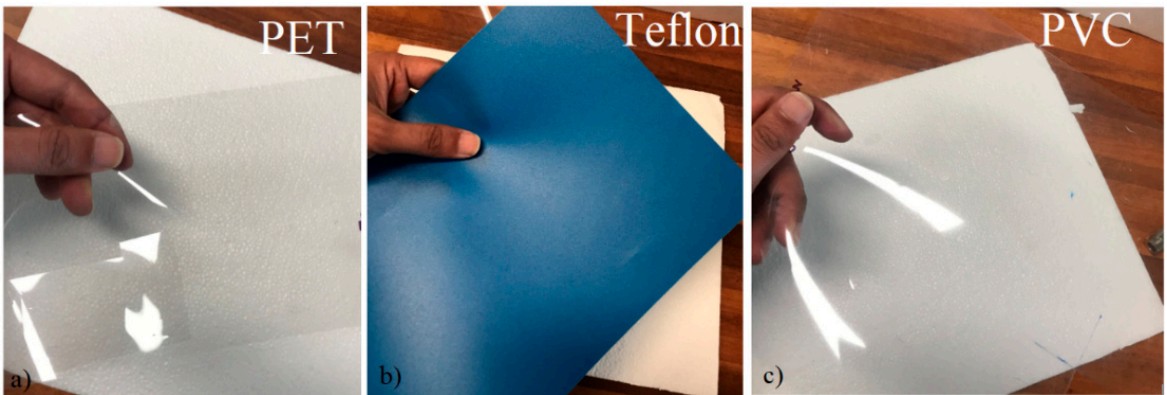

**Figure 4.** Polymer film sheets: (**a**) transparent PET, (**b**) PTFE Teflon, (**c**) transparent PVC.

### 3.3. Antenna Design

Once the comprehensive examination and verification of the properties of the selected materials are completed, the design and the shape selections for the antenna remain crucial. According to different specific applications and frequency requirements, various designs have been previously published for flexible microstrip antennas, their specific applications, and frequency requirements including aperture coupled antennas [22–24], planar inverted-F antennas [25,26], monopole and dipole antennas [27–29], various alphabetic shaped, E-shaped, F-shaped, and H-shaped antennas [14,30–32], and antennas based on the CPW feed [33–35]. These are all microstrip antennas and fabricated on flexible substrates.

The basic microstrip patch antennas are mostly used in wireless applications due to their simple shapes, light weight, and small size, and because they support linear and circular polarization, are capable of dual and triple frequency band operations, have low fabrication and manufacturing costs, and are easy to fabricate and integrate with IoT [11,36–42]. Being extremely compatible with flexible polymer-based antennas, a microstrip patch antenna design was selected for flexibility tests.

Although there are some limitations, such as a narrow bandwidth as a function of the thickness of the substrate material at higher frequencies above 5 GHz, these do not impact our target purposes, including the testing of the bending capabilities of polymers on the radiation characteristics of the antenna. There are, however, also some disadvantages, such as low gain and efficiency, and surface wave excitation [43]. Unfortunately, this surface wave excitation results in power loss as the power is scattered when the antenna bends, thus causing degradation of the signal strength. The low efficiency of the microstrip patch antennas is due to the high $Q$ factor, resulting in these losses in relation to the antenna [44].

Amongst the various shapes possible for microstrip patch antennas, the E-shape is simpler to design and optimize, and was selected herein. The gap between the feedline and the patch $G_{pf}$ (see Figure 5) provides a wider bandwidth compared to a simple patch antenna. Therefore, to determine the flexible polymer-based antenna behaviours for different applications, nine E-shaped microstrip antennas were designed to operate in three frequency ranges: 2.2–2.5 GHz, 2.5–5.0 GHz and 5.0–30 GHz.

To corroborate the impacts of bending, an E-shaped microstrip patch antenna was initially designed on the PET substrate for the operating frequency of 2.45 GHz. Two more antennas with the same shape and characteristics were then made on the Teflon and PVC substrates.

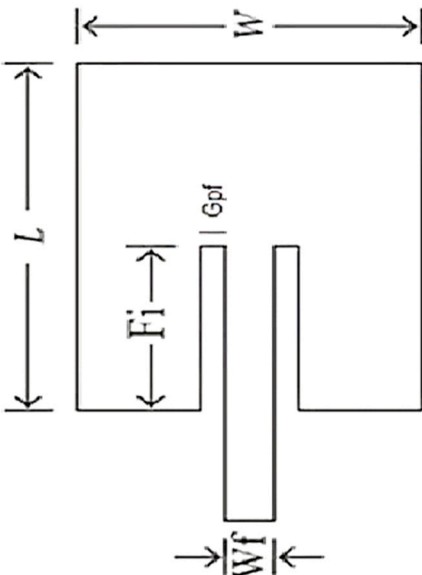

**Figure 5.** Structural diagram of E-shaped microstrip antenna.

### 3.4. Antenna Fabrication

Following the excellent results obtained in the simulation process using CST microwave studio suite 2019, the antenna design was ready for fabrication. For this next step, to fabricate the antenna on the PET substrate, copper was used as the conducting material and the ground plane. An RP-SMA female miniature connector with the dimensions 1.7 × 4.1 mm, 8 mm was used for the microstrip feed line. The thickness of the ground plane and substrate was 0.105 mm and the total thickness with the copper patch was 0.14 mm. A Keysight Vector Network Analyzer (VNA) series E5063A was used to perform the measurements; see Figure 6.

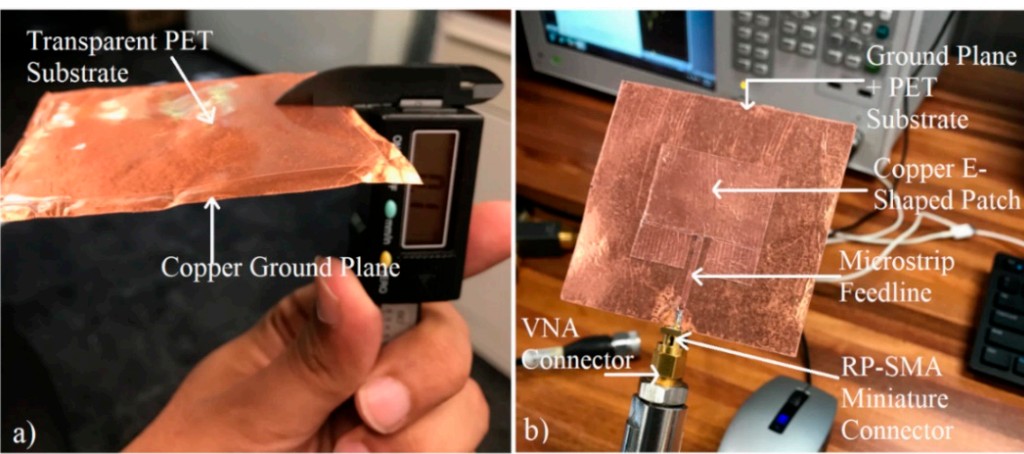

**Figure 6.** E-Shaped flexible microstrip patch antenna on PET substrate: (**a**) thickness measurement, (**b**) fabricated antenna connected to VNA.

Operating at the 2.45 GHz frequency, and following the same procedures and fabrication steps to corroborate the bending effects of radiation on other polymer substrates, the antenna design was then reproduced on both the Teflon and PVC substrates with 100 and 110 μm thicknesses, respectively, as shown in Figure 7.

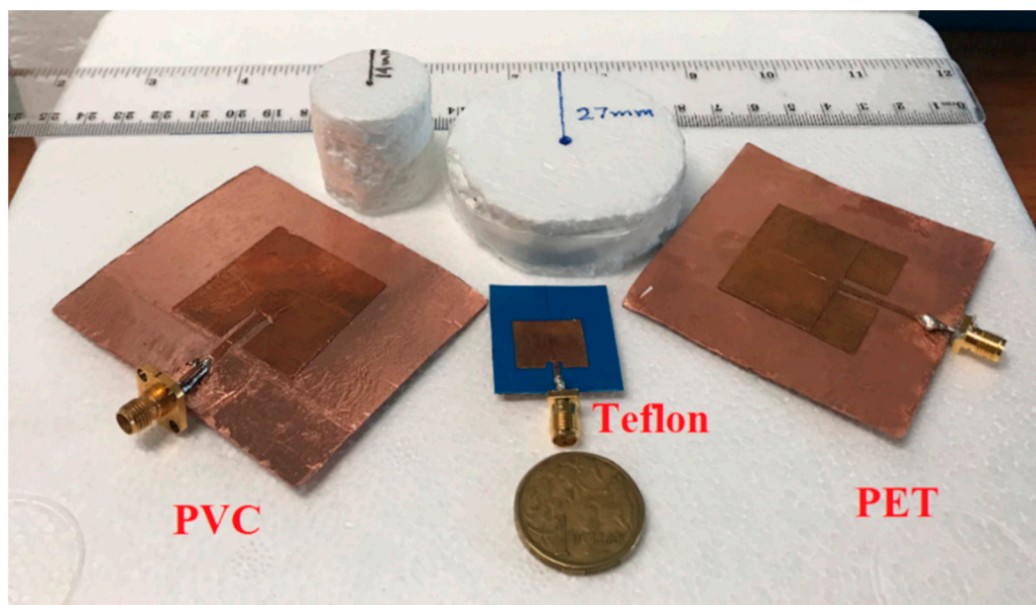

**Figure 7.** Actual aspects of fabricated E-Shaped microstrip antennas operating at 2.45 GHz on substrates PVC (**left**), Teflon (Centre), and PET (**right**).

*3.5. Measurements*

Thus far, this investigation studied the antenna operating at 2.45 GHz. The frequency range analysis was then extended to the three target frequency ranges—2.2 to 2.5 GHz, 2.5 to 5.0 GHz, and greater than 5.0 GHz—to analyze the effects of bending levels of 14 and 27 mm on the three selected antenna substrates—PET, Teflon, and PVC. These three substrates were chosen for their excellent physical, electrical, and thermal properties, as described in Table 2, which shows Teflon has relatively low deformability but is very efficient in moist conditions and has high thermal efficiency, and PVC has very high flexibility but moderate electrical properties. Moreover, PET and PTFE Teflon are commercially available in the very cost-effective form of film sheets. Subsequently, nine flexible patch antennas were fabricated on PET, Teflon, and PVC substrates, with three for each range of frequencies, to corroborate the bending impact on the resonant frequency and reflection coefficient, and the resultant impact on the loss tangent variation on the resonant frequency. Correspondingly, three antennas were designed to operate at 2.45 GHz, three at 4.25 GHz, and three at 7.45 GHz.

### 3.5.1. Introduction to the Frankonia Chamber

To conduct the measurements, testing was performed in a Frankonia anechoic chamber with a frequency range of 30 MHz to 40 GHz at a measuring distance of 3.0 m. The external dimensions of the chamber were 7355 × 3755 × 3300 mm and the size of the uniform area was 1.5 × 1.5 m. The chamber provides excellent RF-Shielding, of less than 90 dB for the frequency range of 1–40 GHz, which is typical for the pan-type module made of 2.0 mm galvanized steel; see Figure 8 VNA was used inside the Frankonia anechoic chamber.

### 3.5.2. Measurements of the *S*-Parameters

As discussed previously, the effects of bending flexible polymer-based antennas have an impact on radiation characteristics such as resonant frequency, reflection coefficients, and gain. The impact on the *S*-parameters, which indicates the percentage shift, or the deviation, of the resonant frequency and the loss of signal strength, is discussed in our recent investigation on polymer based flexible antennas in [1]. The reflection coefficient $S_{11}$ of the three polymer substrate antennas operating at 2.45 GHz is presented in Figure 9.

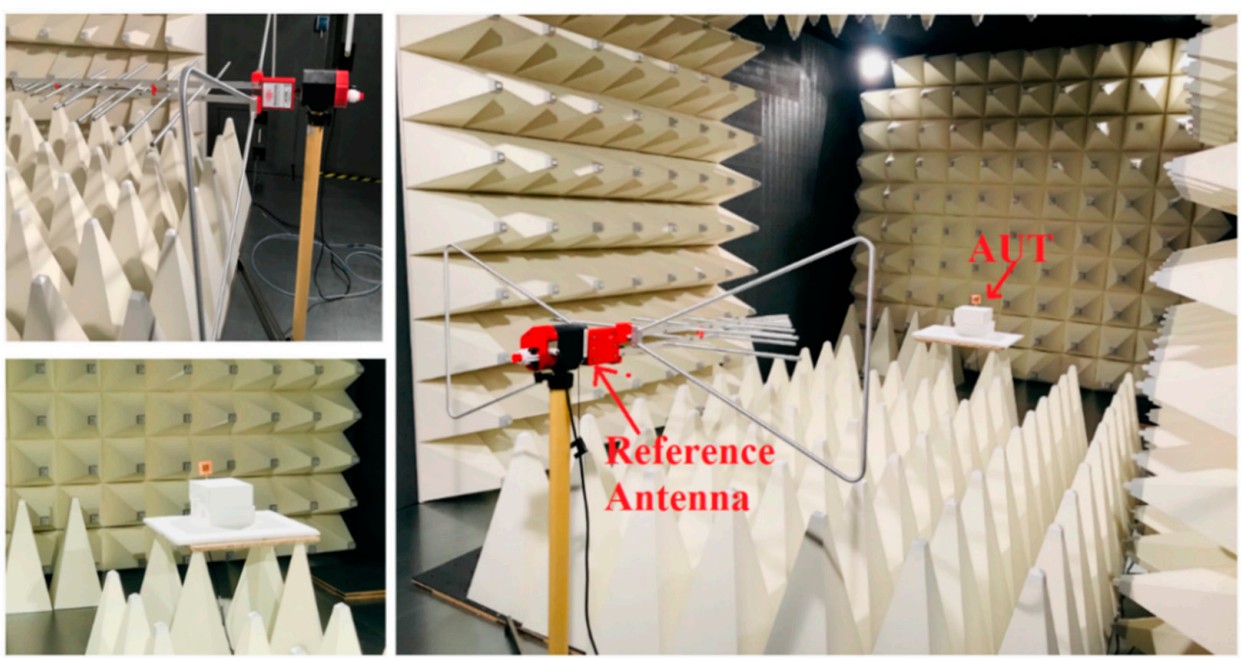

**Figure 8.** Testing arrangements in the Frankonia anechoic chamber.

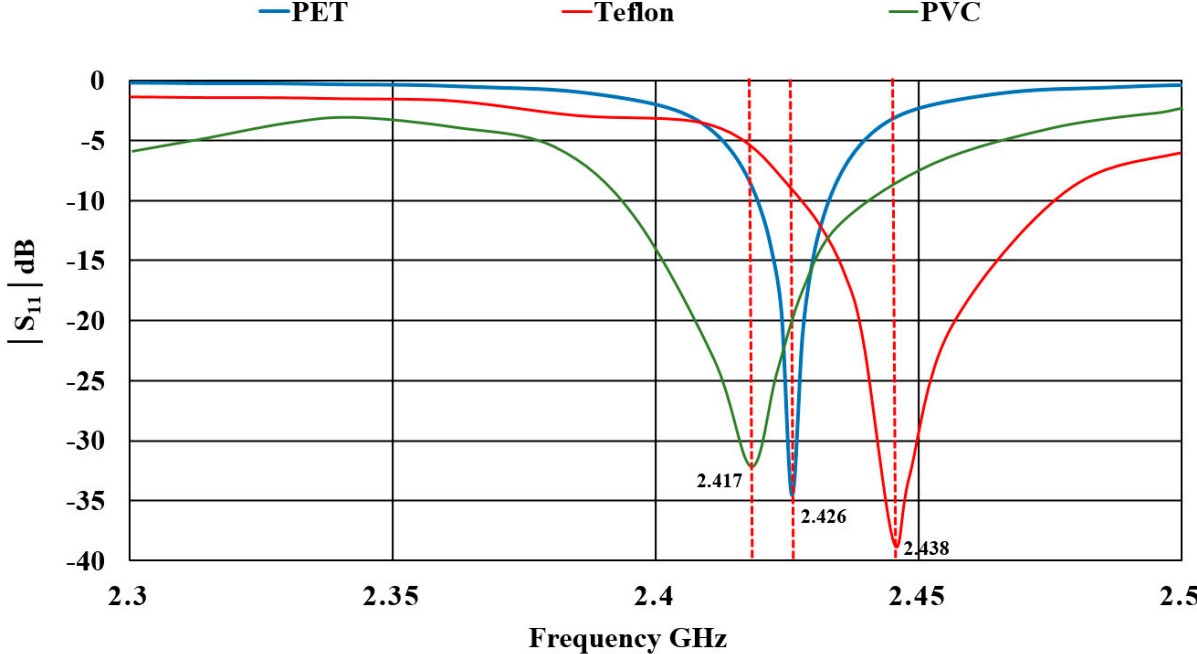

**Figure 9.** Reflection coefficients of E-shaped flexible polymer substrate antennas on PET, Teflon, and PVC substrates operating at 2.45 GHz.

The measured results of the reflection coefficients $S_{11}$ of the PET, Teflon, and PVC substrate antennas operating at 2.45 GHz in the chamber show the resonant frequencies of 2.426, 2.438, and 2.417 GHz, respectively. The reflection obtained for the PET-based antenna is −35 dB, the Teflon-based antenna is −39 dB, and the PVC-based antenna is −32.5 dB. Similarly, the $S_{11}$ parameters of the antennas designed for the operating frequency of 4.25 and 7.45 GHz are presented in Figures 10 and 11, respectively.

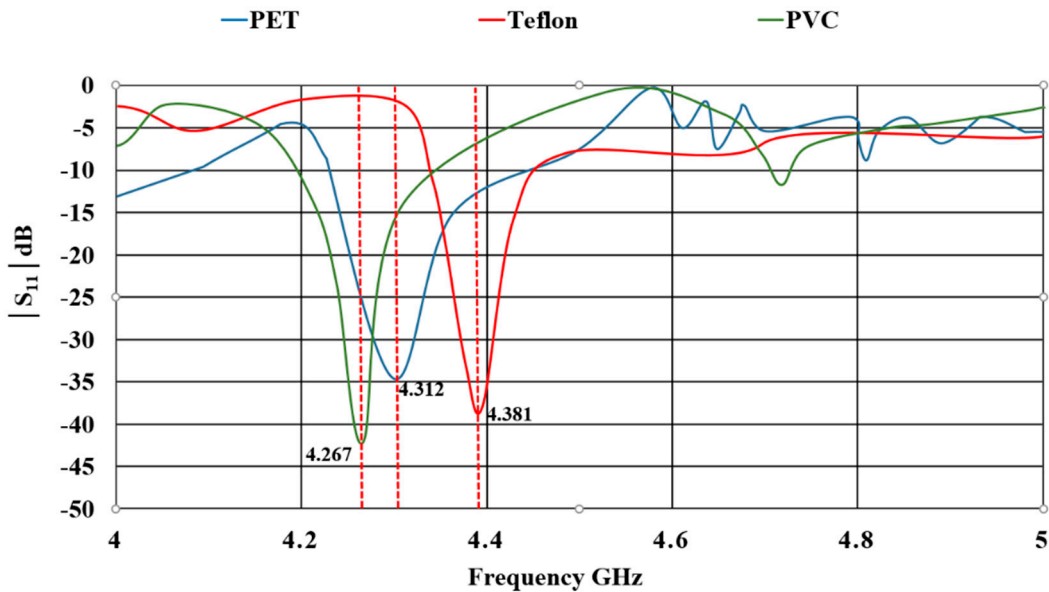

**Figure 10.** Reflection coefficients of E-shaped flexible polymer substrate antennas on PET, Teflon, and PVC substrates operating at 4.25 GHz.

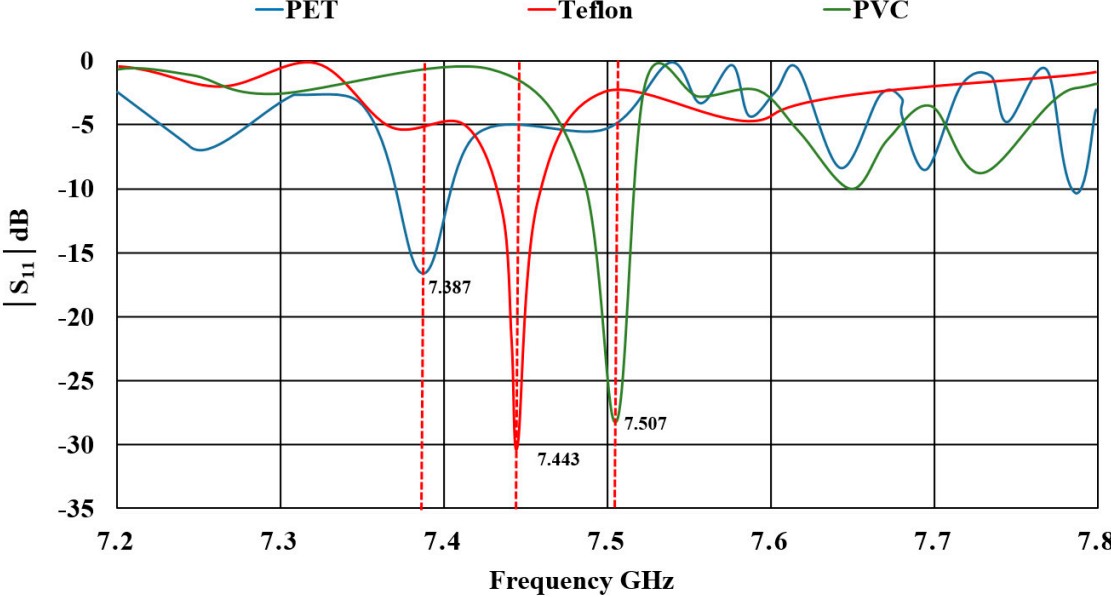

**Figure 11.** Reflection coefficients of E-shaped flexible polymer substrate antennas on PET, Teflon, and PVC substrates operating at 7.45 GHz.

### 4. Bending Analysis

The bending of flexible antennas is different for the different range of frequencies, and becomes crucial at higher frequencies because of the small size of the antenna, impedance mismatching, and the narrow bandwidth. The tested frequency ranges were divided into three groups to analyze the stable response in each range of frequencies, as described in [1]. For this purpose, the effects of bending levels on the PET, Teflon, and PVC substrate antennas for the bending levels of 14 and 27 mm were analyzed for the flexible polymer antennas with an operating frequency of 2.45, 4.25, and 7.45 GHz, which lie within the three frequency ranges of (i) 2.2–2.5 GHz, (ii) 2.5 to 5.0 GHz, and (iii) greater than 5 GHz. To establish a consistent bend in the flexible antenna, polystyrene foam cylinders were used with the radial curvatures of 27 and then 14 mm, as demonstrated in Figure 12. In

this figure, the bending of the flexible antenna on the foam at 27 mm, then 14 mm, and the connection with the VNA are exhibited.

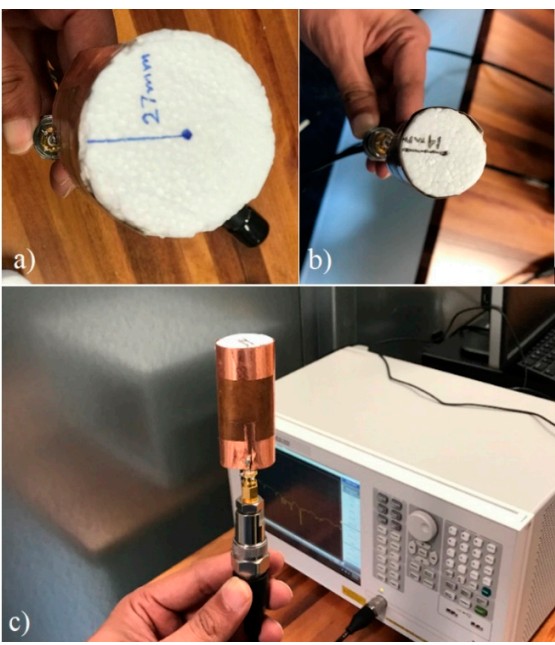

**Figure 12.** Cylindrical polystyrene foam to produce flexible antenna bend (**a**) at 27 mm, (**b**) at 14 mm, (**c**) connected to VNA at 14 mm.

The $S_{11}$ parameters were obtained for these bending stages and compared with the parameters in a flat condition. Figure 13 depicts the reflection coefficients of the flexible PET, Teflon, and PVC substrate antennas operating in the first range at 2.45 GHz with a bending level of 27 mm and then 14 mm. Similarly, the reflection coefficients of each of the flexible antennas operating in the second two ranges, at 4.25 and 7.45 GHz, with bending levels of 27 mm and then 14 mm, are presented in Figures 14 and 15, respectively.

### 4.1. Effect of Bending on Resonant Frequencies

The results obtained from the experimental verifications of the *S*-parameters inside the chamber were analyzed with the shift in the resonant frequency for the various bending levels provided in Table 3, which shows that the bending or curvature with bend states of 27 and 14 mm on polymer-based antennas generate different impacts on their FS.

**Table 3.** Percentage (%) FS in resonant frequencies for flexible antennas on PET, PTFE, and PVC substrates at 3 operating frequencies with bend conditions of 27 and 14 mm.

| | Substrates | | PET | PTFE | PVC |
|---|---|---|---|---|---|
| Resonant Frequency (GHz) | Operating at 2.45 GHz | Flat | 2.426 | 2.438 | 2.417 |
| | | 27 mm | 2.412 | 2.469 | 2.394 |
| | | Shift (%) | −0.58 | 1.25 | −0.96 |
| | | 14 mm | 2.402 | 2.484 | 2.366 |
| | | Shift (%) | −0.99 | 1.85 | −3.42 |
| | Operating at 4.25 GHz | Flat | 4.312 | 4.381 | 4.267 |
| | | 27 mm | 4.442 | 4.392 | 4.294 |
| | | Shift (%) | 2.92 | 0.25 | 0.62 |
| | | 14 mm | 4.468 | 4.453 | 4.366 |
| | | Shift (%) | 3.49 | 1.61 | 2.26 |

**Table 3.** *Cont.*

| Substrates | | | PET | PTFE | PVC |
|---|---|---|---|---|---|
| Resonant Frequency (GHz) | Operating at 7.45 GHz | Flat | 7.387 | 7.443 | 7.507 |
| | | 27 mm | 7.425 | 7.429 | 7.541 |
| | | Shift (%) | 0.51 | −0.01 | 0.45 |
| | | 14 mm | 7.464 | 7.421 | 7.658 |
| | | Shift (%) | 1.03 | −0.29 | 1.97 |

The negative sign with the frequency shift (%) indicates that the shift occurs towards the lowest components of the frequency.

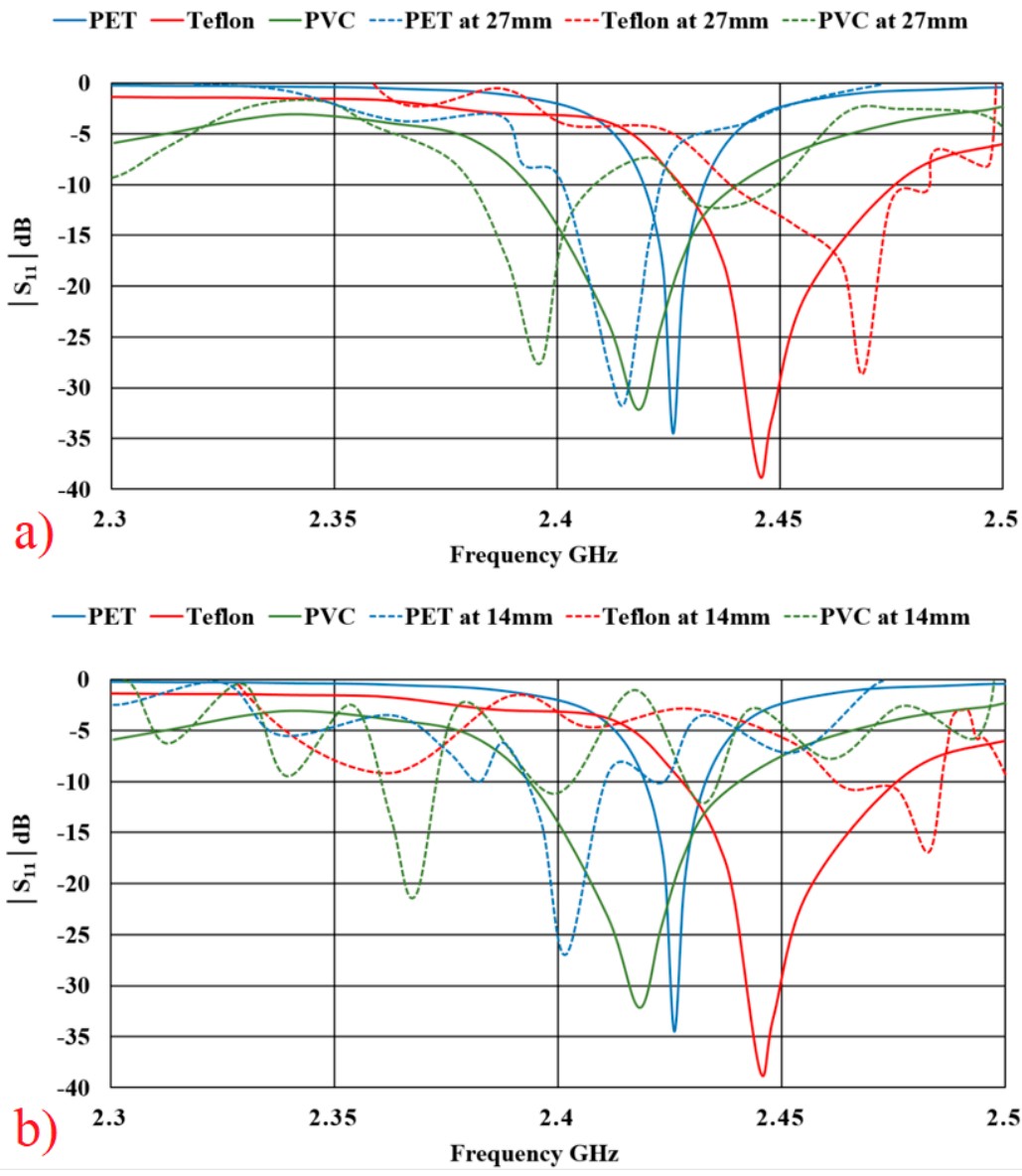

**Figure 13.** Operating at 2.45 GHz, reflection coefficients of E-shaped flexible polymer substrate antennas on PET, Teflon, and PVC substrates non-bent and with bending: (**a**) 27 mm, (**b**) 14 mm.

As observed on the flexible antennas operating at 2.45 GHz, which is a frequency in the ISM-band used for domestic purposes that lies in the first group ranging from 2.2 to 2.5 GHz, the Teflon-based antennas undergo the highest percentage shift in frequency, of approx. 1.25%, towards the highest components of the resonant frequency for a bending state of 27 mm. This result is matched with our investigation in [1]. In contrast, the PET-

and PVC-based flexible antennas possessed less than 1% of frequency shift towards the lowest components of the operating frequency, indicated by the negative sign with the percentage shift. In addition, at the high level of the radial curvature up to 14 mm, the PET substrate antenna yields very little deviation from the resonant frequency compared to the flat state, which is about 0.99% towards the lowest component; see Figure 13.

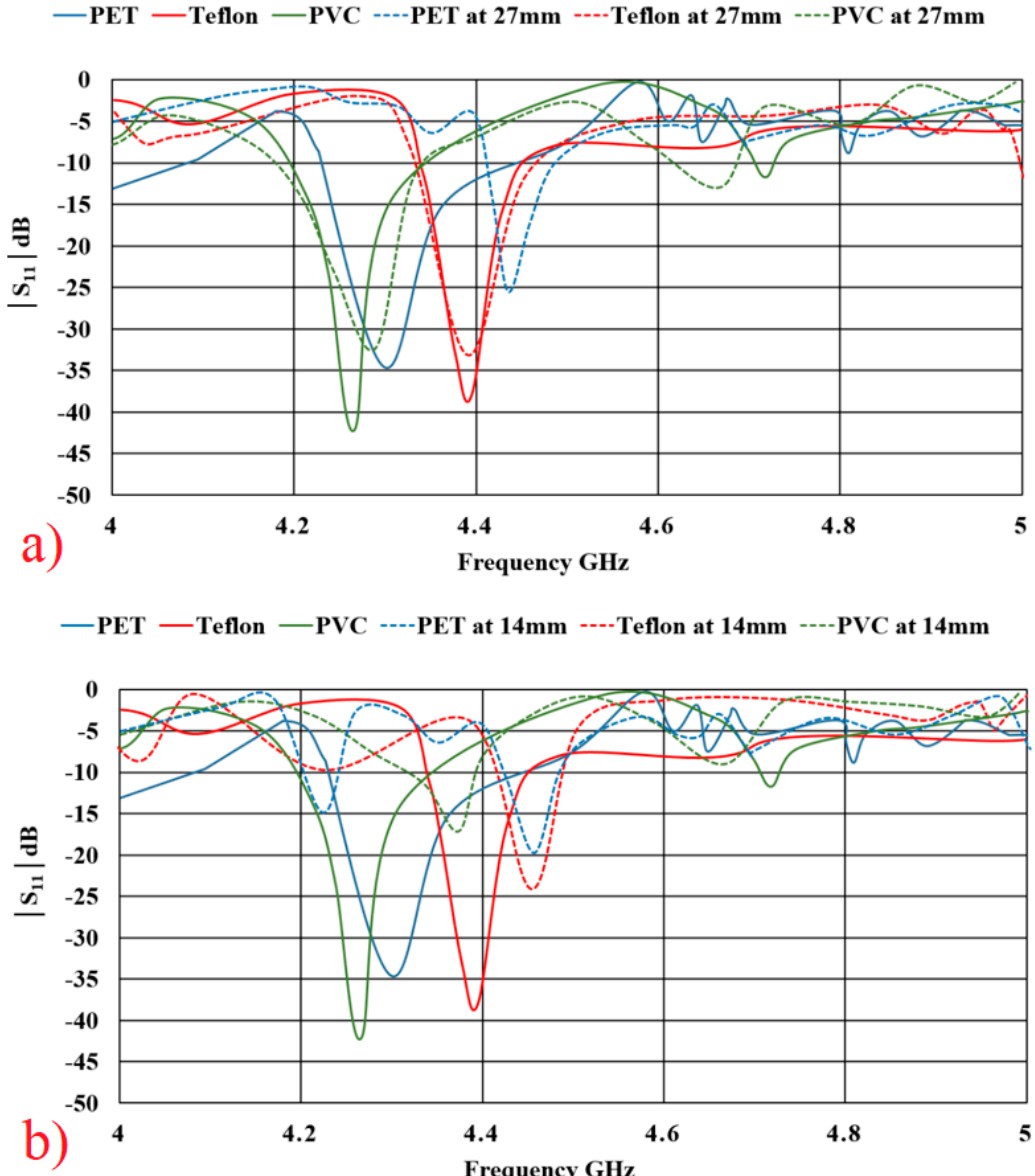

**Figure 14.** Operating at 4.25 GHz, reflection coefficients of E-shaped flexible polymer substrate antennas on PET, Teflon, and PVC substrates non-bent and with bending: (**a**) 27 mm, (**b**) 14 mm.

For the second operating frequency of 4.25 GHz which lies in the C-band suitable for the WiMAX frequency, the *S*-parameters of the flexible microstrip patch antennas reveal that Teflon is much less affected by bending up to 14 mm, providing a 0.25% and 1.61% frequency shift toward the highest components for the bending states of 27 and 14 mm, respectively; see Table 3 and Figure 14. In the first part of this study, in the analysis for the operating frequency range of 2.5–5.0 GHz, PTFE-based flexible antennas were also ascertained to be more efficient by providing 4% of the shift on average. Hence, for this operating range, Teflon is a good candidate as a flexible substrate in terms of the deviation from the resonant frequency of the antenna. Conversely, the PET-based flexible antenna,

which was very efficient within the 2.2–2.5 GHz range, is highly impacted by bending in terms of frequency shift while operating at the 4.25 GHz frequency; see Table 3.

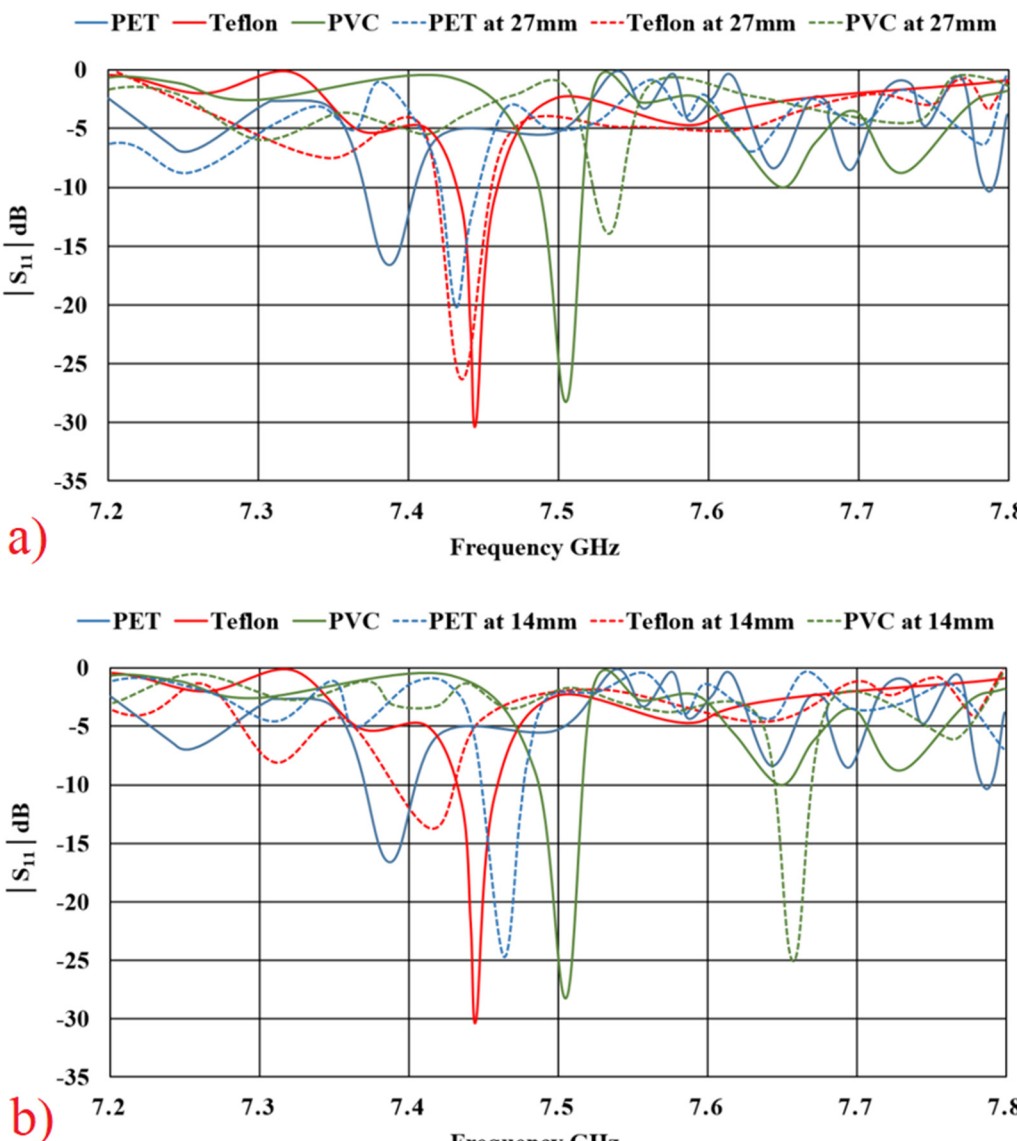

**Figure 15.** Operating at 7.45 GHz, reflection coefficients of E-shaped flexible polymer substrate antennas on PET, Teflon, and PVC substrates non-bent and with bending: (**a**) 27 mm, (**b**) 14 mm.

The decrease in the size of the antenna with the increase in operating frequency and impedance mismatching may be a problem because, with higher level bending and higher frequency levels, the dissipation factor and dielectric constant of the polymer substrates may also vary. Keeping all these conditions in mind and by performing extensive electrical property tests, three antennas were designed and tested in the chamber for the higher operating frequency of 7.45 GHz. At the higher frequencies, above 5 GHz, impedance matching and feed line connection is a critical phenomenon that affects performance. However, although examined and successfully attained, these were not included in the investigative focus on bending capabilities for this experimental study. The PVC-based flexible antenna is impacted most by the highest bending level of 14 mm, which yields a 1.97% shift towards the highest component of frequency, whereas the PET-based substrate antenna gives very little deviation from the resonant frequency compared to the flat case.

### 4.2. Effect of Bending on Reflection Coefficients S11

The *S*-parameters of the designed antennas were critically analyzed to examine the effects of bending on impedance mismatch or the reflection coefficient $S_{11}$ and the signal strength after bending. The impact of bending on the reflection coefficient of the flexible antennas operating at a different range of frequencies was examined by testing the three categories of (i) 2.2–2.5 GHz, (ii) 2.5–5.0 GHz, and (iii) greater than 5.0 GHz, where the antennas, designed on PET, PTFE, and PVC, were analyzed in the chamber for bending levels of 27 and 14 mm. The *S*-parameters depict the impact of the different levels of bending on the reflection coefficient in terms of signal strength and the impedance mismatch with the TL. A significant change in signal strength is caused by the bending or twisting of a flexible antenna because of the impedance mismatch with the TL or the creation of the surface waves. Although signal strength may improve after bending because of its impact on the directivity of the antenna, most of the time it actually decreases; see Figures 13–15.

In Figure 13, when a radial curvature of 27 mm is applied over the flexible antennas operating at 2.45 GHz, compared with the flat status of the antennas, the Teflon-based antenna is highly impacted, with approx. 28% signal degradation, whereas the PET-based antenna is less impacted, with approx. 7.7% reduction in its signal strength. This degradation, although more extreme, is similar to that observed with the 14 mm bending process, where the Teflon-based antenna suffered a high 55% reduction in signal strength, and the PET-based antenna demonstrated the least degradation with 23%.

For the experimental analysis of the flexible antennas operating at 4.25 GHz, however (Figure 14), the Teflon-based antenna suffered the least degradation. Compared with the flat status of the antennas, for the bending level of 27 mm, the Teflon substrate bore the least impact of only 14% of signal strength reduction, which simultaneously defines its impedance mismatch at 27 mm and confirms the analysis in [1]. The signal strength reduction for both the PET and PVC substrate antennas is almost similar at approx. 22%. Indeed, with the 14 mm bending, the PVC-based flexible antenna suffered an immense signal reduction of approx. 60%, whereas the Teflon substrate antenna possesses the least signal strength degradation of 25% lower than its flat status strength.

In Figure 15, which compares the signal strengths in a flat condition for the flexible antennas operating at the higher 7.45 GHz frequency, the *S*-parameters illustrate that for the bending of the antennas to 27 mm, Teflon, as the substrate of the antenna, shows the least degradation with a low value of 12%. This contrasts with the PVC-based antenna, which is heavily impacted, with an approx. 50% signal reduction. However, when the bending curvature increases to 14 mm, the impact measurements are reversed, with the Teflon substrate antenna being impacted with a high 55% reduction in its signal strength. The experimental results demonstrate that, with around 9% reduction, the PVC-based antenna has the highest tolerance for the increased bending, with its comparatively low signal strength degradation and impedance mismatch being the least affected in this category. In this analysis, a PVC-based antenna was ascertained to be the most efficient for the frequency range greater than 5 GHz.

### 4.3. Dielectric Constant (ε) and the Resonant Frequency Shifts (%)

The permittivity $\varepsilon$ of the polymer substrate, its dielectric constant, is a complex value that depends on the frequency, the roughness of the surface of the material, and the temperature. As mentioned in [1], flexible substrates for wearable applications normally have dielectric constants from 2 to 12, where lower values of its dielectric constant, from 2.2 to 3.5, serve to reduce the surface waves and increase the bandwidth and gain of the antenna in frequencies ranging from 2.2 to 7.5 GHz. Consequently, the three substrates used for the flexible antennas in this study—PET, Teflon, and PVC—have the dielectric constants of 2.07, 2.70, and 3.70, respectively.

For the flexible antennas operating at 2.45 GHz, the PET-based flexible antenna, which possesses a dielectric constant of 2.07, just under the ideal range of 2.2 to 3.5, is the least impacted by bending at 14 mm with a shift of only 0.99%. By comparison, with the flexible

antennas operating at 4.25 and 7.45 GHz, Teflon, with its dielectric constant inside the ideal range of 2.7, proves to have a low impact in terms of the resonant frequency deviation from its flat status for the highest level of bending up to 14 mm. It is observed that the Teflon-based antennas, for the operating frequencies of 4.25 and 7.45 GHz, possess shifts from the resonant frequency of only 1.61% and 0.29%, respectively. By comparison, the PVC-based flexible antennas, with their 3.7 dielectric constants, which are above the ideal range, possess a high shift in frequency while undergoing bending at 14 mm, where they provide the highest level of deviation of 3.42% and 1.97% for the operating frequencies of 2.45 and 7.45 GHz, respectively. Hence, from these experimental verifications, in addition to the comprehensive literature review in [1], a modified range of dielectric constants of 2–3.5 is recommended for polymer substrate antennas operating in a 2.2–7.5 GHz range for flexible wearable applications.

## 5. Conclusions

A practical implementation of flexible antennas with polymer substrates PET, PTFE, Teflon, and PVC polymer substrates operating at 2.25, 4.25, and 7.45 GHz, is presented in this paper and compared with the evaluations of our comprehensive study of polymer-based antennas in [1]. Due to the rapid growth of IoT and subsequent interconnection of various smart devices, the flexibility of these IoT's is an important aspect which, in turn, requires flexible antennas for communication. Therefore, an experimental approach to examine bending capabilities of polymer substrate flexible antennas is presented in each of the different range of frequencies, whereby the effects of bending levels of 27 and 14 mm on the PET, PTFE Teflon, and PVC substrate antennas were analyzed. For the antennas operating in the first range at 2.45 GHz, the PET-based and PVC-based flexible antennas were observed to possess a less than 1% frequency shift towards the lowest components, with the PTFE (Teflon)-based antennas showing the highest impact with a percentage shift of frequency of 1.25% for the 27 mm bend state. With the 14 mm radial curvature, the PET substrate antenna gave the best performance with 0.99% of frequency deviation from its flat status. For the second range, at an operating frequency of 4.25 GHz, the Teflon-based antenna contrastingly possessed the lowest frequency deviation, which was 0.25% and 1.61% for bending levels of 27 and 14 mm, respectively. For the operating frequency of 7.45 GHz, in the third range, the Teflon-based flexible antenna presented almost no impact of bending at 27 mm, and for 14 mm only a 0.29% shift towards the lowest components of the operating frequency. The PVC-based flexible antenna was impacted the most for the highest bend level where, in terms of the impedance mismatch at 14 mm bending, with a 1.97% shift, the PVC substrate antenna was ascertained to be the most efficient for this frequency range. Hence, this successful bending analysis of polymer-based flexible antennas proved their reliability in terms of bending for various stages of curvature, and provided a means to select a suitable polymer substrate for future wearable sensors and antennas with high bendability without significantly distorting the radiation characteristics. Due to their bendability and flexibility characteristics, these polymer-based antennas are advantageous for incorporation in designs for future smart sensors, including applications in the Internet of Things (IoT).

**Author Contributions:** Conceptualization, M.U.A.K., R.R. and F.T.; methodology, M.U.A.K.; software, M.U.A.K.; validation, R.R., P.I.T. and F.T.; formal analysis, M.U.A.K.; investigation, M.U.A.K. and F.T.; resources, P.I.T.; data curation, M.U.A.K.; writing—original draft preparation, M.U.A.K.; writing—review and editing, R.R., F.T., P.I.T.; visualization, M.U.A.K.; supervision, R.R. and F.T.; project administration, R.R.; funding acquisition, R.R. and F.T. All authors have read and agreed to the published version of the manuscript.

**Funding:** This research received no external funding.

**Institutional Review Board Statement:** Not applicable.

**Informed Consent Statement:** Not applicable.



**Data Availability Statement:** Not applicable.

**Conflicts of Interest:** The authors declare no conflict of interest.

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
