# Peer review of "The Impact of Bending on Radiation Characteristics of Polymer-Based Flexible Antennas for General IoT Applications"

_applsci, doi:10.3390/app11199044_

Round 1

Reviewer 1 Report

Unfortunately, this work looks like a student thesis. In the Introduction section the well known information is provided.

Only one type of antenna was tested. Results show that tested antennas are quite narrowband, clearly not suitable for 2.4GHz Wi-Fi, which would require at least 5 times wider bandwidth to support all Wi-Fi channels. Using wide band antenna most likely would eliminate any bending effects.  

Quality of photos is poor. The figures are thoughtless and not arranged to meet the requirements of a scientific journal.

 I do not want to offend authors, but this publication, especially in this version, is not suitable for any scientific journal.

Author Response

We would like to thank the reviewer for his/her review and consideration. We took all comments into account and revised the paper accordingly. In this revised version, we took into account the suggested comments and improvements. We addressed them point by point as reported in this reaction letter.

Reviewer 2 Report

In this study, the bending capabilities of flexible polymer substrate antennas for general IoT applications are practically analysed by fabricating flexible antennas on various polymer substrates. I have few comment below.

  1. They can start their introduction stating the wearable devices.
  2. There could be additional reference that they can add in terms of wearable devices and mechanical testing of flexible devices. e.g. 1) Facile fabrication of paper-based silver nanostructure electrodes for flexible printed energy storage system, Materials & Design 151, 1-7. 2) "Machine Washable Conductive Silk Yarns with a Composite Coating of Ag nanowires and PEDOT:PSS", ACS Applied Materials & Interface, 2020, 12, 24, 27537–27544. 3)"Effect of Varying the Density of Ag Nanowire Networks on Their Reliability During Bending Fatigue", Scripta Materialia, 2019, 161, 70-73. 4)Stretchable and patchable composite electrode with trimethylolpropane formal acrylate-based polymer, Composites Part B: Engineering 163, 185-192

Author Response

(The authors gave the same response as above.)

Reviewer 3 Report

The flow of the paper and the approach should be better explained in the introductory section.

All along the paper the intermediate results should be wrapped up, helping with small tables detailing the performance for each material/frequency.

the final comparison table should be clearly evidenced and commented.

A series of mispells should be corrected along the paper.

Author Response

(The authors gave the same response as above.)

Round 2

Reviewer 1 Report

No progress at all! 

Author Response

Look at the attachment

Reviewer 2 Report

I think the manuscript can be acceptable now.

Author Response

Look at the attachment
